# Cytotoxicity and Genotoxicity Evaluation of *Zanthoxylum rhoifolium* Lam and In Silico Studies of Its Alkaloids

**DOI:** 10.3390/molecules28145336

**Published:** 2023-07-11

**Authors:** Rufine Azonsivo, Kelly Cristina Oliveira de Albuquerque, Ana Laura Gadelha Castro, Juliana Correa-Barbosa, Helena Joseane Raiol de Souza, Andryo Orfi de Almada-Vilhena, Gleison Gonçalves Ferreira, Anderson Albuquerque de Souza, Andrey Moacir do Rosario Marinho, Sandro Percario, Cleusa Yoshiko Nagamachi, Julio Cesar Pieczarka, Maria Fâni Dolabela

**Affiliations:** 1Postgraduate Program in Pharmaceutical Sciences, Federal University of Pará, Belém 66075-110, PA, Brazil; azonsivo@gmail.com (R.A.);; 2Postgraduate Program in Biodiversity and Biotechnology of the BIONORTE Network, Federal University of Pará, Belém 66075-110, PA, Brazil; kellyoalbuquerque@gmail.com (K.C.O.d.A.);; 3Postgraduate Program in Pharmaceutical Innovation, Federal University of Pará, Belém 66075-110, PA, Brazil; lauracastro.farmacia@gmail.com (A.L.G.C.); correabjuliana@gmail.com (J.C.-B.); 4Postgraduate Program in Risk and Natural Disaster Management in the Amazon, Federal University of Pará, Belém 66075-110, PA, Brazil; helena.souza@embrapa.br; 5Center for Advanced Studies of the Biodiversity and Cell Culture Laboratory, Guamá Science and Technology Park, Federal University of Pará, Belém 66075-750, PA, Brazil; andryoorfi@hotmail.com (A.O.d.A.-V.);; 6Faculty of Pharmacy, Federal University of Pará, Belém 66075-110, PA, Brazil; 7Faculty of Chemistry, Federal University of Pará, Belém 66075-110, PA, Brazil

**Keywords:** *Zanthoxylum rhoifolium*, cytotoxicity, mutagenicity, molecular docking

## Abstract

The alkaloids isolated from *Zanthoxylum rhoifolium* have demonstrated great pharmacological potential; however, the toxic profiles of these extracts and fractions are still not well elucidated. This study evaluated the toxicity of the ethanol extract (EEZR) and neutral (FNZR) and alkaloid (FAZR) fractions. Chemical characterization was performed by chromatographic methods: thin-layer chromatography (TLC) and high-performance liquid chromatography coupled with diode array detection (HPLC–DAD). The cytotoxicity of the samples was evaluated in human hepatocellular carcinoma (HepG2) cells using the cell viability method (MTT) and mutagenicity by the *Allium cepa* assay (ACA). Alkaloids isolated from the species were selected for toxicity prediction using preADMET and PROTOX. The molecular docking of the topoisomerase II protein (TOPOII) was used to investigate the mechanism of cell damage. In the EEZR, FNZR, and FAZR, the presence of alkaloids was detected in TCL and HPLC–DAD analyses. These samples showed a 50% inhibitory concentration (IC_50_) greater than 400 μg/mL in HepG2 cells. In ACA, time- and concentration-dependent changes were observed, with a significant reduction in the mitotic index and an increase in chromosomal aberrations for all samples. Nuclear sprouts and a micronucleus of the positive control (PC) were observed at 10 µg/mL and in the FAZR at 30 µg/mL; a chromosomal bridge in FNZR was observed at 105 µg/mL, CP at a concentration of 40 µg/mL, and nuclear bud and mitotic abnormalities in the EEZR were observed at 170 µg/mL. The alkaloids with a benzophenanthridine were selected for the in silico study, as structural alterations demonstrated certain toxic effects. Molecular docking with topo II demonstrated that all alkaloids bind to the protein. In summary, the fractionation of *Z. rhoifolium* did not interfere with toxicity; it seems that alkaloids with a benzophenanthridine nucleus may be involved in this toxicity.

## 1. Introduction

Numerous specimens of plants from the Rutaceae family are used by traditional Brazilian communities for medicinal purposes. The species *Zanthoxylum rhoifolium* Lam., a large tree known in Brazil as mamica-de-cadela, is widely used as an antipyretic [1], for digestion aid, and in the treatment of dyspepsia, flatulence, colic, earaches [2,3], diarrhea, and hemorrhoids [4].

There have been numerous chemical studies of *Z. rhoifolium* isolated alkaloids (Figure 1) with a benzophenanthridine nucleus: bocconoline (**1**), chelerythrine (**2**), 6-acetonyldihydrochelerythrine (**3**), zanthoxyline (**4**), rhoifoline A (**5**), rhoifoline B (**6**), fagaramide (**7**), avicine (**8**), oxyavicine (**9**), nitidine (**10**), and oxynitidine (**11**), and those with a furoquinoline nucleus: skimmianine (**12**) and E-Z-dimethylroifolinate (**13**) [5,6,7,8,9,10,11,12].

Several biological activities of extracts from *Z. rhoifolium* have been described, such as action against promastigotes and amastigotes [13], an inhibitory effect on macrophage infection in *Leishmania amazonensis* [14], activity against intestinal nematodes [15], and antibacterial [7,16], antifungal [17], antiplasmodial [18], and antitumor activities [19]. Moreover, several of these activities have been attributed to alkaloids including antiviral action [20] and neuromuscular blockade [21]. Furthermore, alkaloids with benzophenanthridine rings showed a high cytotoxicity against tumor cells [19,22].

A preliminary toxicity assessment has already been carried out with an ethanolic extract of *Z. rhoifolium* against *Artemia salina*, showing a potential toxic effect for this species [23]. Another study demonstrated that fractionation significantly contributed to the reduction in cytotoxicity in normal gastric cells, since the EEZR was more cytotoxic and the FAZR was moderately cytotoxic, whereas the FNZR, and a possible alkaloid, were not cytotoxic [24]. It is noteworthy that cytotoxicity is not yet well understood as a response to genotoxic agents, because responses to genotoxic damage are complex and can range from repair, damage fixation, mutations, damage deletions, and cell death [25]. Thus, there is a need to investigate the genotoxic potential of the species.

Although certain studies have demonstrated the pharmacological potential of *Z. rhoifolium*, and there are preliminary toxicity studies with conflicting results, there is still a lack of studies that investigate whether this possible toxicity is related to the alkaloids present in this species. The in silico prediction of the toxicity of compounds allows us to choose among the available components those which will progress to future phases of analysis, thus utilizing mathematical methods that, when based on structure-property models, allow a reduction in expenses and avoid the unnecessary use of animals [26]. Given the above, this study evaluated the cytotoxic and mutagenic activity of extracts and fractions from *Z. rhoifolium*, as well as investigating the in silico toxicity of certain alkaloids present in the species and the possible mechanism of cell death.

## 2. Results

### 2.1. Phytochemical Studies

The ethanol extract of *Z. rhoifolium* (EEZR; yield = 15.72%), the fraction of neutrals of *Z. rhoifolium* (FNZR; yield = 9.6%), and the fraction of alkaloids of *Z. rhoifolium* (FAZR; yield = 1%) were submitted to thin-layer chromatographic analysis (TLC), showing suggestive bands of alkaloids.

The EEZR was analyzed by HPLC–DAD, wherein the chromatogram suggested the presence of high-to-medium polarity compounds. The peak with the highest intensity (RT = 2.00 min) presented a UV spectrum with a λmax of 221.5, 274.7 nm (Figure 2A). The peak at 3.00 min presented a UV spectrum with a λmax of 226.3, 270.0 nm (Figure 2B), and the peak at 32.00 min presented a UV spectrum with a λmax of 219.2, 265.3 nm (Figure 2D), both of which are suggestive of the indole ring present in an indole alkaloid. The peak at 7.00 min showed UV bands also suggestive of indole alkaloids, whose UV spectrum displayed a λmax of 226.3, 271.2, and 310.4 nm. In the chromatograms of FNZH (Figure 2) and FAZH (Figure 2), there is a peak with a close retention time and a UV spectrum similar to the extract (Figure 2B).

### 2.2. Toxicity of Z. rhoifolium

#### 2.2.1. Cytotoxicity Assays

In the assessment of cytotoxicity for the HepG2 strain, the EEZR, FNZR, and FAZR seemed to have a low cytotoxic potential and their IC_50_ values were greater than 400 µg/mL, suggesting that the exposure time of the cells did not influence the toxicity (Figure 3).

#### 2.2.2. Allium Cepa Test

For the positive control, EEZR, FNZR, and FAZR, at all concentrations, there was a decrease in the mitotic index (MI) in a time-dependent fashion. Regarding the chromosomal aberration index (AI), there was an increase in this index with increasing exposure time for all samples (Table 1).

For the EEZR, the cell-division phase was influenced by the concentration. At the concentration of 340 µg/mL, in the 24 h analysis, the cells were found predominantly in anaphase and telophase, while at the intermediate concentrations (170 µg/mL; 85 µg/mL; and 42.5 µg/mL) for the same period, there was a greater predominance of cells in prophase and metaphase, and at the concentration of 21.25 µg/mL, in prophase. A similar trend was observed for the FNZR and FAZR samples. The phase of cell division predominant for the FAZR was influenced by time; that is, at all concentrations, cells were predominantly in metaphase (Figure 4, Table 1). For the FNZR, the phase of the cell was influenced by the concentration; at 420 µg/mL, cells were found to be in prophase and anaphase (Figure 4 and Table 1).

For AI, the EEZR was influenced by concentrations of 40 µg/mL and 170 µg/mL in metaphase, and for the FNZR, in anaphase stage, by concentrations of 105 µg/mL and 52.5 µg/mL, revealing nuclear-bridge abnormalities and a nuclear bud (Figure 5C).

For the positive control, a greater mutagenicity profile was revealed when compared to the test samples (Figure 5 and Table 1).

#### 2.2.3. In Silico Studies

In order to evaluate the participation of alkaloids in the toxicity of the extract and fractions, in silico studies were performed. To select the molecules included in this study, a broad literature review of the species was carried out; thus, 29 metabolites isolated from *Z. rhoifolium* were selected. All compounds are alkaloids isolated and identified from this species and the biological activities were related to this class (Figure 5).

Initially, all molecules underwent toxicity analysis for algae, microcrustaceans, and fish, in which all substances were designated as toxic, except for the compound bis[6-(5,6-dihydrochelerythrinyl)] (**23**). In regard to the evaluation of the crustacean *Daphnia* sp., the substance magnoflorine (**7**) was considered non-toxic.

The toxic events of the isolated substances demonstrated many differences between the molecules, revealing a vast diversity of components. For the evaluation of toxicity in *Medaka* fish, the tests revealed substances such as nitidine (**10**), syncarpamide (**16**), and bis[6-(5,6-dihydrochelerythrinyl)] (**23**) that have a limited toxic capacity. All of the other substances caused highly toxic events for both Medaka and Minnow fishes. Despite their toxic potential for fish, several alkaloids did not seem to cause mutagenic effects (**1**, **5**, **7**, **10**, **11**, **12**, **13**, **14**, **15**, **16**, **17**, **23**, and **26**); furthermore, several of these alkaloids did not show a carcinogenic potential in mice or rats (**3**, **5**, **7**, **12**, **13**, **15**, **16**, **17**, **20**, **2**, and **29**; Table 2). Compounds **2**, **3**, **4**, **6**, **9**, **10**, **18**, **19**, **20**, **21**, **22**, **24**, **25**, **27**, **28**, and **29** appear to possess a mutagenic potential. While only Compound **19** caused mutagenicity in the four selected strains of *Salmonella typhimurium:* TA100_10RLI, TA100_NA, TA1535_10RLI, and TA1335_NA, the other compounds caused mutagenicity in only one to three of the four strains. As for cytotoxicity assessment, all of the molecules appear to have a weak-to-moderate inhibitory action on the Herg channel (Table 2).

In addition, the compounds were evaluated with a program that allows the prediction of in vivo effects, with the most promising molecule being Compound **29**. Regarding the prediction of acute oral toxicity, Compounds **3**, **13**, **17**, **20**, and **21** proved to be toxic if ingested. The other molecules proved to be harmful if ingested. Only Compounds **1**, **2**, **3**, **4**, **5**, **6**, **7**, **8**, **9**, **11**, **12**, **13**, **14**, **15**, **16**, **17**, **18**, **20**, **21**, **22**, and **25** displayed immunotoxic effects. However, Compounds **19**, **24**, **27**, **28**, and **29**, despite no toxic effects being detected, proved to be harmful when ingested (Table 3).

### 2.3. Molecular Docking

Initially, a redocking was performed using the co-crystallized ligand (ANP) with the topo-II enzyme (PDB code: 1ZXM) to evaluate the molecular docking parameters of the AutoDock program, which should be satisfactory to describe the macromolecular target and the ligand conformation. The root-mean-square deviation (RMSD) value between the experimental and docked (ANP) values was 0.58 Å, as determined using the Discovery Studio Visualizer (Dassault Systèmes BIOVIA, Discovery Studio Modeling Environment, Release 2017, San Diego: Dassault Systèmes, 2016). When RMSD values were less than 2 Å, the model achieved satisfactory solutions and reproduced the orientation of the co-crystallized ligand [26]. In this way, the model demonstrated a good ability to predict the orientation of the crystallographic ligand in the active site of the topo-II enzyme. Figure 6 demonstrates the interactions between the ligands and the macromolecule.

The structural affinity parameters demonstrated that rhoifoline A had the lowest free energy ΔG (Table 4). The intermolecular interactions of the receptor–ligand compound in the active site of topo II were observed according to Table 4. The intensity of intramolecular hydrogen bonding can be affected by the nature, size, and position of the substituents that can induce different effects in the delocalization of π electrons and hydrophobic and π bonds—important characteristics favorable to interactions with the substrate-binding cavity of topo II.

## 3. Discussion

Chemical studies of the extract and fractions of *Z. rhoifolium* have detected the presence of alkaloids, especially indole alkaloids, for which the literature demonstrates their antiparasitic potential [27,28,29].

In the present study, the EEZR, FNZR, and FAZR presented values suggestive of a relatively low cytotoxicity (IC_50_ > 400 μg/mL), indicating that the exposure time and the increase in concentration did not interfere in cell viability, as shown in Figure 3 These results differ from the study conducted by Barbosa (2022) using VERO cell lines, presenting IC_50_ values for the EEZR and FAZR of 330.6 μg/mL and 111.7 μg/mL, respectively, considered moderately cytotoxic. However, the FNZR remained in the low cytotoxicity range (831.9 μg/mL) [13].

The cytotoxicity of the extracts was also evaluated using the ethanolic extract from the stem bark of *Z. rhoifolium* by determining the cytotoxic concentration (CC_50_) against murine peritoneal macrophages. The results indicated that the sample may be moderately or not cytotoxic, displaying a CC_50_ > 400 µg/mL [14].

Another study demonstrated that fractionation contributed significantly to the reduction in cytotoxicity in normal gastric cells, since the EEZR was more toxic (IC_50_ = 77.08 µg/mL), the FAZR was moderately cytotoxic (IC_50_ = 127.5 µg/mL), and the FNZR, and a possible alkaloid, were not cytotoxic (IC_50_ > 500 µg/mL) [24]. In summary the neutral fraction and alkaloid fraction showed a low cytotoxicity in normal gastric cells, but there is a lack of toxicity studies on this species in the literature, what reinforces the importance of this study.

Genotoxicity was observed for the EEZR, FNZR, and FAZR. Another study evaluated the toxicity of the EEZR and its hexane fraction (FHZR), with the EEZR presenting cytotoxicity by inhibiting growth and the mitotic index in the *Allium cepa* assay, in addition to increasing the frequency of chromosomal and micronucleus aberrations. On the other hand, in an in vivo study with bone marrow cells, this alteration was not observed. FHZR does not have a reduced toxic potential [25].

In the present study, the EEZR showed a genotoxic potential, but the aim of the fractionation was to obtain fractions with higher alkaloid contents, the result was different from the previous study [25], and the extract and fractions were toxic. Certain alkaloids are electrophilic and highly reactive, and they can interact with proteins resulting in hepatotoxic, genotoxic, mutagenic, and carcinogenic effects, such as DNA cross-linking, DNA–protein cross-linking, and chromosomal aberrations [27].

Studies have already demonstrated the toxic potential of alkaloids isolated from plants, an example being camptothecin (CPT) isolated from *Campthoteca acuminata*, which has a high antitumor activity, and its use being limited due to its high toxicity [28]. CPT is a potent inhibitor of nucleic-acid synthesis and an inducer of single-strand breaks in mammalian DNA [29]. Therefore, the premise of this work was that alkaloids may be involved in the genotoxicity of *Z. rhoifollium*.

Therefore, in order to evaluate the toxicity of alkaloids, in vitro and in vivo methods can be employed, with the need to isolate and identify alkaloids in sufficient quantity to carry out these tests. These challenges combine to make the evaluation time-consuming and costly. In the screening for toxicity, in silico assays can be used, which are easy to perform, fast, inexpensive, have a good reliability [30], and do not require isolation of the molecule.

The premise of this search was that the genotoxic potential of the EEZR and its fractions is related to alkaloids; therefore, prediction studies were carried out on metabolites belonging to this class. Dihydrobenzophenanthridine-type alkaloids were isolated from *Z. rhoifolium*. In the literature, these alkaloids were both active and inactive against bacteria, and there were significant variations in cytotoxicity [31,32]. Therefore, dihydrobenzophenanthridine-type alkaloids were selected, including 11 alkaloids of this class.

Another class of alkaloids isolated from *Z. rhoifolium* is the quinoline alkaloids that have demonstrated antimicrobial activity, as described. The alkaloids (+/−)-6-Acetonyldihydrochelerythrine and avicine were active against strains of *Staphylococcus aureus*, *S. epidermidis*, *Streptococcus pyogenes*, *Escherichia coli*, and *Klebsiella pneumoniae* [11]. In another study that evaluated the antibacterial activity of alkaloids, the alkaloids 6-acetonyldihydronitidine, 6-acetonyldihydroavicine, and zanthoxyline also displayed activity against strains of *S. aureus*, *S. epidermidis*, *Micrococcus luteus*, *K. pneumoniae*, *Salmonella setubal*, and *E. coli* [7].

In a toxicity study with the leaf oil of *Z. rhoifolium* against *Artemia salina*, the oil presented an LD_50_ of 18.5 μg/mL, showing a potential toxic effect for this species [33]. In order to compare which class has the highest toxic potential, we included alkaloids from both classes. In silico studies using seaweed, microcrustaceans, and fish showed that most alkaloids belonging to both classes were toxic or very toxic (Table 3). The test using the crustacean *Artemia salina* is a simple test, considered adequate for analysis of the toxicity of chemical substances [34], as well as others such as *Daphnia magna* [35], but additional toxicity studies are important.

It is possible that a high concentration of a certain chemical compound has an effect (inhibitory or stimulatory) on the cell cycle, such as the effect of caffeine in *Drosophila prosaltans* [36], mefloquine on human lymphocytes [37], extracts of *Alpinia mutans* and *Pogostemun heyneanus* on the root cells of *Allium cepa* [38], and glauocolide B extracted from *Vemonia eremphila* Mart. on human lymphocytes [39].

The *Allium cepa* test showed that all of the samples raised the mitotic index but this reduced with increasing exposure time. Mitotic indices significantly lower than those of the negative control may indicate changes arising from the actions of chemical substances in the growth and development of the exposed organisms, and MIs greater than the negative control result from the increase in cell division, which may be harmful to cells, leading to disordered cell proliferation and, eventually, the formation of tumors [40].

Unlike the MI, the number of chromosomal aberrations grew over time, significantly differing from the NC (*p* < 000.5). The observed chromosomal aberrations are mainly the nuclear bud, characterized as a genomic error where there is a failure in the DNA repair process or further expansion, and nucleoplasmic bridges, which occur by the deletion of unbalanced rearrangements of cell division [41].

In this sense, all of the tested samples, regardless of concentration, demonstrated clastogenic effects on the *Allium cepa* system. The increase in the number of CA, even with the decrease in the MI, suggests that the extract and fractions of *Z. rhoifolium* present a greater genotoxic effect than a cytotoxic one, and that the alkaloids found in the species are the main components responsible for this mechanism, promoting the stabilization of the mitotic spindle, which causes a greater number of chromosomal aberrations to occur, as seen during in silico studies.

The results of the Ames test, performed in silico using the PreADMET tool, indicated 22 molecules with mutagenic potential. However, only 4 of these compounds also showed the potential to promote carcinogenesis in rats and mice, using the in silico study carried out with the same tool. From this potential mutagenic effect, a compound can potentially induce carcinogenicity, since mutagenic agents can alter the sequence of DNA bases causing an error in the division process that leads to the perpetuation of that cell. It should be noted that the Ames test is based on mutation induction in *Salmonella typhimurium* and not in rodents [42].

All molecules analyzed, except for Compounds **22**, **25**, **30**, and **37**, did not demonstrate full carcinogenic potential, a favorable safety aspect for them. The importance of in silico carcinogenicity assessment is emphasized, since carcinogenicity can be a critical aspect in the development of drugs. This aspect, if not properly managed, can lead to suspension or delays in drug approval by regulatory authorities [43].

In predicting acute oral toxicity, a wide range of LD_50_ values was observed, with non-toxic and very toxic alkaloids (Table 3). Several factors can influence the acute oral toxicity of compounds, such as their physical–chemical characteristics that can influence absorption, bioavailability, and distribution [44].

## 4. Materials and Methods

### 4.1. Plant Material and Chemical Studies

The barks of *Z. rhoifolium* were collected in May 2019, in the city of Belém—Pará (Brazil), at the Empresa Brasileira de Pesquisa e Agropecuária (EMBRAPA), and a specimen was deposited in the IAN herbarium (Instituto Agronômico do Norte, number: 199947). The plant material was washed in running water, dried in a forced air oven (40 °C), grounded, and the powder subjected to exhaustive maceration in 96° GL ethanol for 7 days, with daily stirring 3 times a day.

The extracted solution was concentrated in a rotary evaporator (T= 40–45 °C) until residue, originating the ethanolic extract (EEZR). The EEZR was subjected to acid–base partition, yielding the neutral (FNZR) and alkaloid (FAZR) fractions. All samples were submitted to analysis by thin-layer chromatography (TLC), using silica gel as the stationary phase and a mobile phase of dichloromethane: methanol: ammonium hydroxide, in the proportion of 85:15:0.2, using Dragendorff reagent and UV developer (365 nm) [45].

Samples were characterized by high-performance liquid chromatography (Waters 1525), with a DAD detector (Waters 2998), in a stationary phase composed of a C18 column (Sunfise c18 4.6 × 150 mm, 5 µm) and a mobile phase with a linear gradient composed of deionized water plus 0.1% formic acid (eluent A) and acetonitrile plus 0.1% formic acid (eluent B). The oven temperature remained at 40 °C, the injection volume used was 20 µL, and the flow rate was 0.5 mL/minute. The readings were taken in a UV–DAD detector at wavelengths of 210–600 nm, registering chromatograms at wavelengths of 215, 229, 280, 290, and 320 nm.

### 4.2. Cell Viability Assay with Tetrazolium Salt (MTT)

The human hepatoma cell strain HepG2 was used, cultivated in Roswell Park Memorial Institute 1640 medium (RPMI 1640; Gibco) supplemented with 10% fetal bovine serum (SBF, Gibco) and stored in a 5% CO_2_ incubator at 35 °C in a humid atmosphere (UltraSafe). The cells underwent weekly passages, periodically checking the morphological characteristics and presence of microorganisms.

The EEZR, FNZR, and FAZR were solubilized in distilled water, followed by successive dilutions to the following concentrations: 400 μg/mL; 200 µg/mL; 100 µg/mL; 50 µg/mL; 25 µg/mL; and 12.25 µg/mL. For this purpose, 96-well plates were used in which HepG2 cells were plated (1 × 10^4^ cells/mL) and incubated at 37 °C in a humid atmosphere with 5% CO_2_.

After 24 h of incubation, treatment with the EEZR, FNZR, and FAZR was carried out and readings were performed at 24 h and 48 h of treatment. After this period, MTT (5 mg/mL) was added and incubated for 4 h. DMSO (dimethyl sulfoxide) was then added after 1 h, and the absorbances of the wells were read in a multiple-well scanning spectrophotometer (SARSTEDT) at 490 nm [46].

### 4.3. Mutagenic Activity through the Allium Cepa Test

Five concentrations were used in each sample, alongside distilled water as a negative control, and colchicine as a positive control. *A. cepa* seeds were placed to germinate in Petri dishes, lined with filter paper moistened with each sample. When the roots reached about 1 cm, they were incubated for 24, 48, and 72 h before fixing with Carnoy for 24 h, followed by washing in distilled water (3 × /5 min). The roots were immersed in hydrochloric acid to 1M for 15 min for hydrolysis, followed by 3 more washes in distilled water (5 min each). The roots were stained with 2% acetic orcein for a period of 10 min [47].

For each sample, 2 to 3 roots were placed on a slide and 1 or 2 drops of orcein were added, using a scalpel to cut the final end of the root (meristematic region), discarding the rest. The same tissue was covered with a cover slip, ensuring the pressure carefully to maintain an even surface of the material placed between the slide and cover slip. The edges of the cover slips were sealed with nail polish to prevent the material from drying out. Finally, cells were visualized and read under the microscope with a 10× objective to observe the material and at a 40× magnification to observe the mitotic index and the parameters for mutagenic analysis.

The mitotic index (MI) and the aberration index were evaluated as: nuclear bud (A), micronuclei (B), chromosome bridge (C), nuclear fragmentation (D), and mitotic abnormalities (E), while the mitotic index and aberration index were determined by calculations.

All tests were performed in triplicate, and statistical analysis was performed using the chi-square test of the BioEstat 6.0. During analysis, *p* values of less than or equal to 0.05 were considered statistically significant. For the analysis of normality, the Lilliefors test and the Kruskal–Wallis test were used.

### 4.4. In Silico Studies of Alkaloids

Initially, a bibliographical survey of preexisting studies of *Z. rhoifolium* was carried out, where its isolated chemical components were selected. A total of 43 compounds were chosen, with 11 alkaloids belonging to the benzophenanthridine class and the remaining alkaloids belonging to the quinolinic class. To carry out the drawing of chemical structures, the ChemSketch program (12.1.0.31258) was used, saving in an appropriate format (.mol) (Mcule-2019).

Toxicity was evaluated using the PreADMET program, adopting the following criteria: algae: <1 mg/L considered toxic and >1 mg/L non-toxic; *Daphnia*: <0.22 µg/mL considered toxic and >0.22 µg/mL non-toxic; *Medaka* and *Minnow* fish: <1 mg/L considered very toxic, 1–10 mg/L toxic, 10–100 mg/L harmful, and >100 mg/L non-toxic. For the prediction of mutagenicity, the Ames test was used, being evaluated for different strains of *Salmonella typhimurium*. Regarding the carcinogenic capacity, mouse and rat models were adopted [48,49,50,51].

For the oral toxicity study, the ProTox-II online server [52] was used, based on the 50% lethal dose (LD_50_), following the classification of this potential into the following classes: fatal if ingested—class I (LD_50_ ≤ 5mg/kg) and class II (LD_50_ between 5 and 50 mg/kg); toxic if ingested—class III (LD_50_ between 50 and 300 mg/kg); harmful if ingested—class IV (LD_50_ between 300 and 2000 mg/kg); may be harmful if ingested—class V (LD_50_ between 2000 and 5000 mg/kg); non-toxic—class VI (LD_50_ ≥ 5000 mg/kg) [53].

Cytotoxicity, hepatotoxicity, acute toxicity, and immunotoxicity were also evaluated, both for the nuclear stress response and binding to nuclear receptors. Regarding biological activities, the Pass online server was used for its prediction, with the exclusion criteria of a probability of activity (Pa) greater than 0.7 (>70%) [54].

### 4.5. Molecular Docking

Chemical structures were prepared from SMILES files, downloaded from the Pubchem database [55]. The crystallographic structure of the topo-II enzyme was retrieved from the Protein Data Bank (PDB) under the code 1ZXM [56] with a resolution of 1.87 Å. Molecular docking simulations were performed using the AutoDock v.4.2 program, wherein the binding affinity of the receptor–ligand complex was estimated using a scoring function that calculates the change in free energy after binding [57].

All water molecules were removed, polar hydrogen atoms were added, and the atom charges of proteins and ligands were assigned by Kollman and Gasteiger methods. The center of the grid box from 1ZXM for AutoDock was at x: 36.503, y: −0.165, z: 37.843, a size of 60 × 60 × 60 Å points, and a spacing of 0.375 Å. Molecular docking calculations were performed considering the scoring function of the Lamarckian Genetic Algorithm [56]. Binders performed 10 interactive runs and the fit to the best scoring result was considered for the calculation of mean square deviation (RMSD). Analyses of intermolecular interactions and RMSD calculations were performed using the Discovery Studio Visualizer (Dassault Systèmes BIOVIA, Discovery Studio Modeling Environment, version 2021, San Diego: Dassault Systèmes, 2021). RMSD values below 2 Å showed that the redocking was successful according to literature data [57].

## 5. Conclusions

The EEZR, FNZR, and FAZR displayed IC_50_ > 400 µg/mL. All samples showed cytotoxic and genotoxic effects, and alkaloids were detected. All alkaloids evaluated by in silico studies showed toxic effects for at least three organisms (algae, crustaceans, and fish). Therefore, regarding mutagenicity, certain molecules showed mutagenic and carcinogenic potential. Very few alkaloids proved to be carcinogenic. Regarding acute oral toxicity, none of the alkaloids proved to be toxic. Regarding immunotoxicity, a considerable portion of the alkaloids studied demonstrated an immunotoxic potential. The alkaloids rhoifoline A, rhoifoline B, zanthoxyline, and decarine stabilized the topo-II–DNA complex.

## Figures and Tables

**Figure 1 molecules-28-05336-f001:**
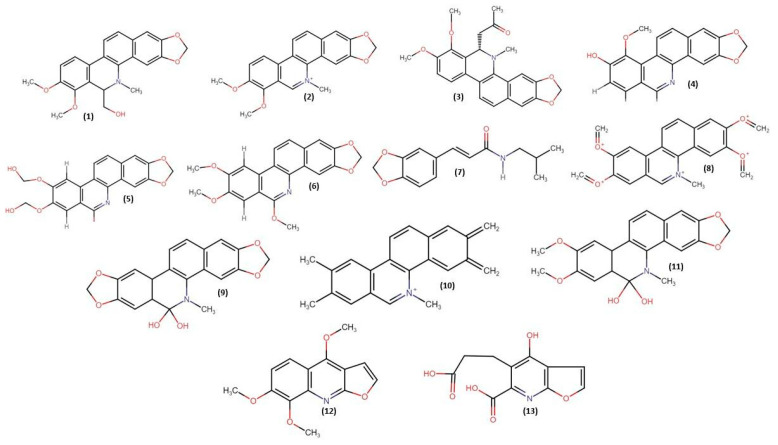
Benzophenanthridine and furoquinolone alkaloids isolated from *Z. rhoifolium*.

**Figure 2 molecules-28-05336-f002:**
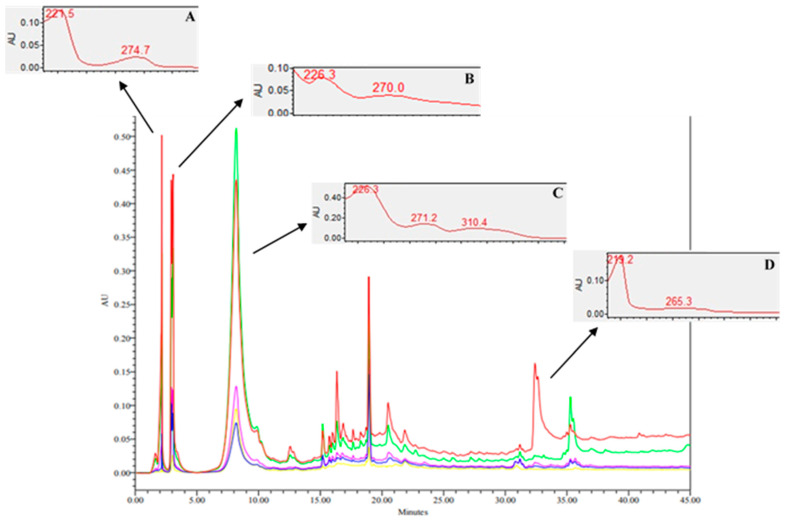
Chromatograms of *Z. rhoifolium* extract and fractions and their ultraviolet spectra. Chromatographic conditions: temperature: 40 °C; flow: 0.5 mL/min; volume: 20 µL; column: C18; mobile phase: deionized water, 0.1% formic acid (eluent A), and acetonitrile plus 0.1% formic acid (eluent B); reading at wavelengths of 215, 229, 290, and 320 nm. Legend: 1—ethanol extract of *Z. rhoifolium* (EEZR); 2—fraction of neutrals (FNZR); 3—fraction of alkaloids (FAZR); A–D: ultraviolet spectra with absorbances suggestive of chromophores present in alkaloids.

**Figure 3 molecules-28-05336-f003:**
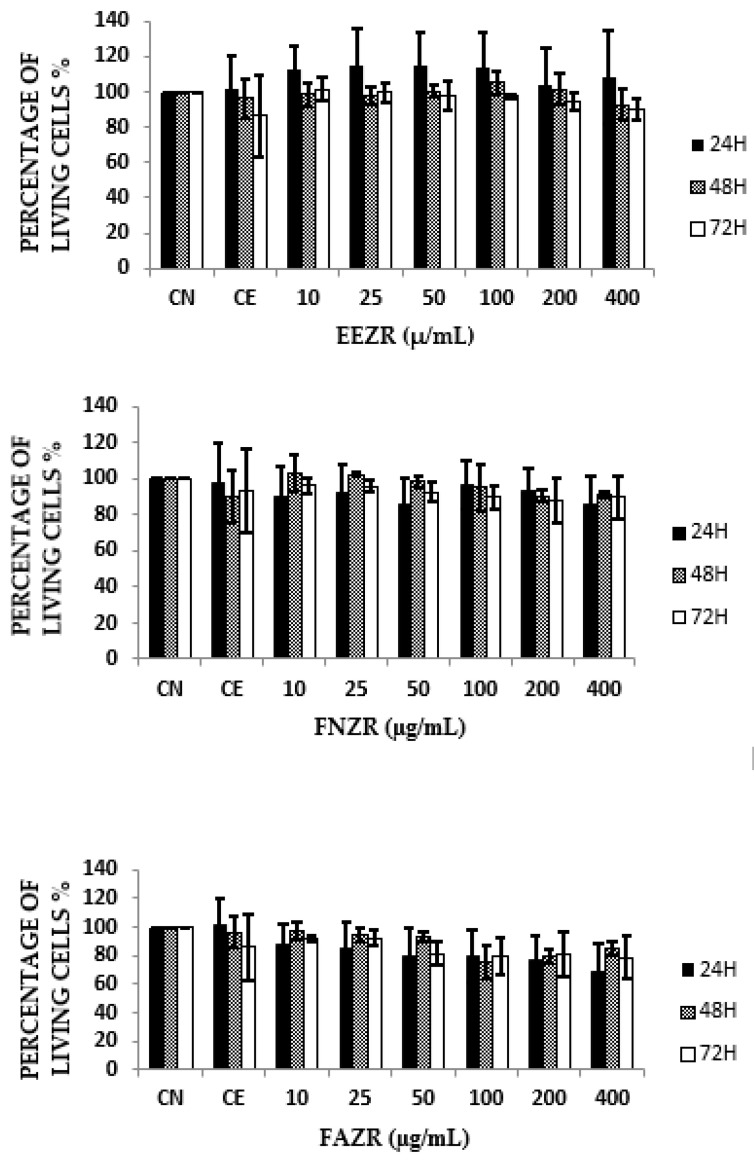
Determination of cell viability at different exposure times and concentrations. EEZR: *Zanthoxylum rhoifolium* ethanol extract; NC: negative control; CE: ethanol concentration; FNZR: neutral fraction of *Zanthoxylum rhoifolium*; FAZR: alkaloid fraction of *Z. rhoifolium*.

**Figure 4 molecules-28-05336-f004:**
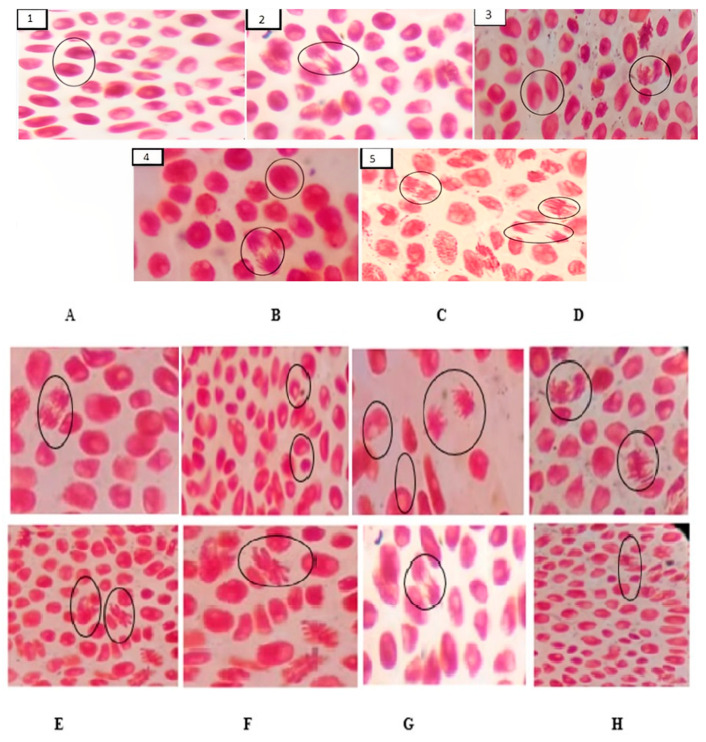
Mitotic index of the samples after 24 h at different concentrations and analysis of the cellular alterations caused by *Z. rhoifolium*. Mitotic index of the samples after 24 h at different concentrations: (**1**): prophase for the NC, telophase for the FAZR; (**2**): prophase and telophase for the PC, telophase and metaphase for the FAZR; (**3**): prophase for the FNZR; (**4**): prophase and telophase for the PC, telophase and metaphase for the FAZR; (**5**): prophase and telophase for the FAZR. Mutagenic effect at different concentrations after 24 h: (**A**): anaphase with a nuclear bridge and a nuclear bud for the NC after 72 h; (**B**): micronucleus and a nuclear bud for the PC at 10 µg/mL; (**C**): PC nuclear abnormalities at the concentration of 40 µg/mL; (**D**): C-metaphase, delay in anaphase at 80 µg/mL after 48 h for the EEZR; (**E**): mitotic abnormalities, T-metaphase for the EEZR at 170 µg/mL; (**F**): C-metaphase and a nuclear bud for the FNZR at 52.5 µg/mL; (**G**): chromosome-bridging anaphase for the FNZR at 105 µg/mL, (**H**): C-metaphase and a nuclear bud for the FAZR at 30 µg/mL.

**Figure 5 molecules-28-05336-f005:**
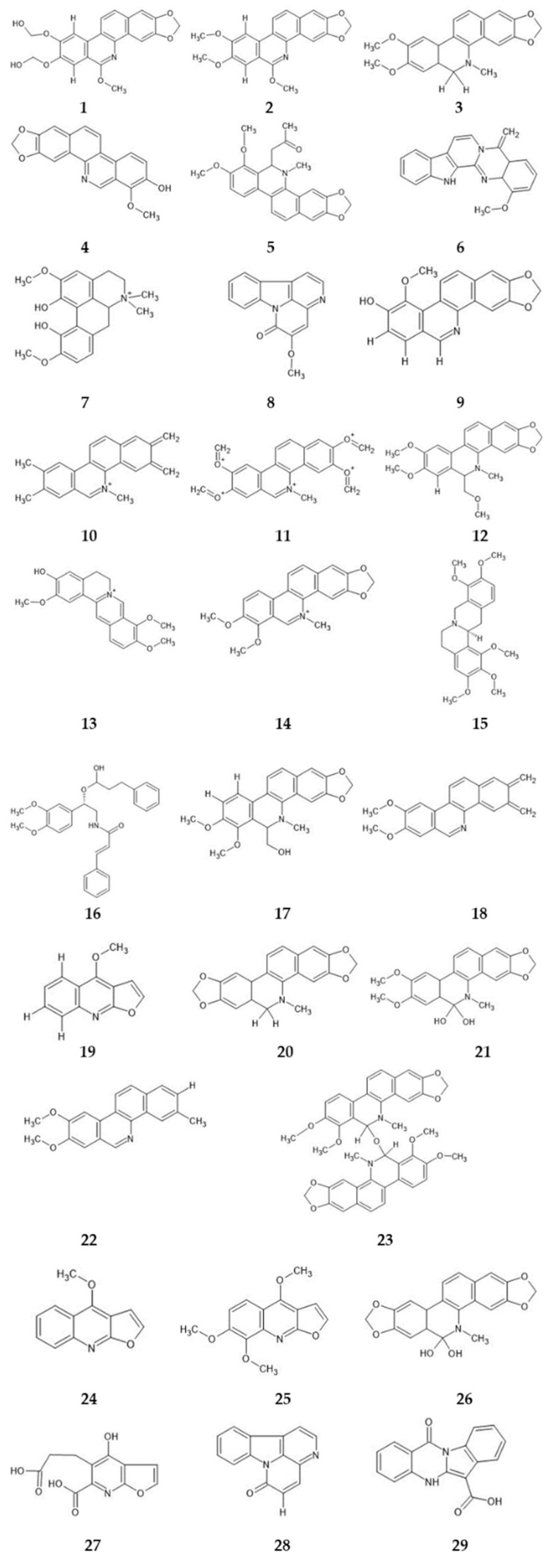
Alkaloids selected for prediction studies: **1**—rhoifoline A, **2**—rhoifoline B, **3**—dihydronitidine, **4**—decarine, **5**— (+/−)-6-Acetonyldihydrochelerythrine, **6**—1-methoxy-7,8-dehydrorutaceacarpine, **7**—magnoflorine, **8**—5-methoxy-canthin-6-one, **9**—zanthoxyline, **10**—nitidine, **11**—avicine, **12**—6-acetonyldihydroavicine, **13**—columbamine, **14**—chelerythrine, **15**—O-methylcapaurine, **16**—syncarpamide, **17**—bocconoline, **18**—nornitidine, **19**—dictamine, **20**—dihydroavicine, **21**—oxynitidine, **22**—norfagaronine, **23**—bis[6-(5,6-dihydrochelerythrinyl)], **24**—dectamnine, **25**—skimmianine, **26**—pelitorine, **27**—Z-dimethylrhoifolinate, **28**—canthin-6-one, and **29**—quinazoline-6-carboxylic acid.

**Figure 6 molecules-28-05336-f006:**
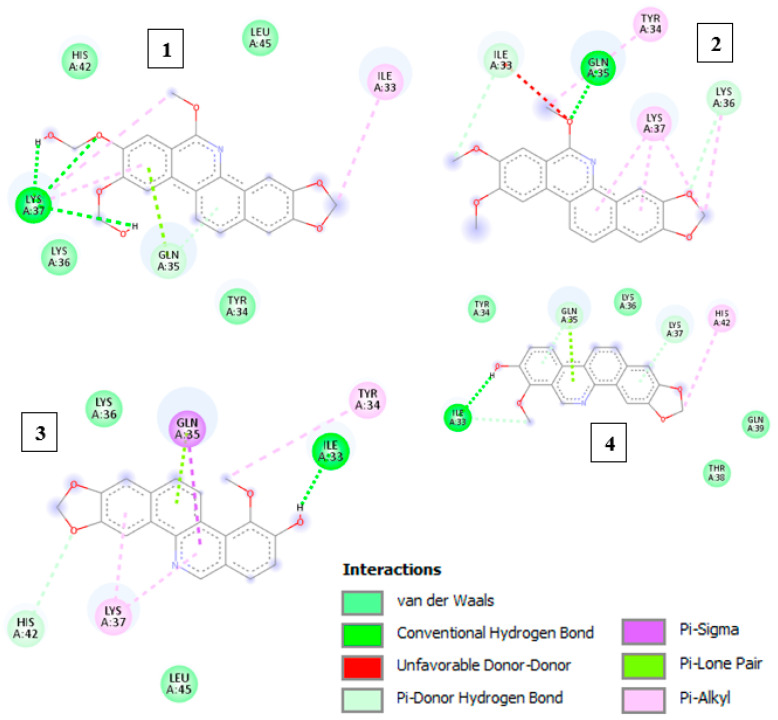
Representations of the intramolecular interactions of the compounds rhoifoline A (**1**), rhoifoline B (**2**), zanthoxyline (**3**), and decarine (**4**).

**Table 1 molecules-28-05336-t001:** Determination of mitotic and aberration indices of cells treated with *Z. rhoifolium*.

Samples (µg/mL)	Mitotic Index (%)			Aberration Index (%)		
	24 h	48 h	72 h	24 h	48 h	72 h
NC	8.5	5.3	3.8	0.06	0.1	0.2
PC						
160	ND	ND	ND	ND	ND	ND
80	38.0 *	28.0 *	20.9 *	0.56 *	1.14 *	1.44 *
40	29.0 *	24.9 *	16.5 *	0.52 *	1.06 *	1.36 *
20	26.3 *	22.0 *	12.1 *	0.48 *	0.96 *	124 *
10	23.6 *	19.9 *	11.9 *	0.42 *	0.88 *	1.14 *
EEZR						
340	37.1 *	29.0 *	19.0 *	0.52 *	0.72 *	1.18 *
170	33.7 *	24.8 *	17.1 *	0.44 *	0.68 *	1.10 *
85	29.0 *	22.3 *	15.0 *	0.40 *	0.64 *	1.00 *
42.5	25.0 *	20.6 *	12.7 *	0.38 *	0.58 *	0.88 *
21.25	24.7 *	18.2 *	10.0 *	0.34 *	0.54 *	0.78 *
FNZR						
840	ND	ND	ND	ND	ND	ND
420	36.7 *	30.0 *	19.0 *	0.46 *	0.78 *	1.18 *
210	31.9 *	25.0 *	15.9 *	0.44 *	0.72 *	1.08 *
105	27.8 *	21.8 *	12.9 *	0.40 *	0.66 *	0.96 *
52.5	24.5 *	18.0 *	10.8 *	0.34 *	0.56 *	0.86 *
FAZR						
120	ND	ND	ND	ND	ND	ND
60	38.0 *	28.9 *	19.6 *	0.38 *	0.70 *	1.12 *
30	30.6 *	25.0 *	16.7 *	0.32 *	0.66 *	1.04 *
15	26.0 *	22.0 *	13.4 *	0.30 *	0.60 *	0.92 *
7.5	23.0 *	19.1 *	11.0 *	0.28 *	0.50 *	0.80 *

NC—negative control; PC—positive control (colchicine); ND—not determined; EEZR—ethanol extract; FNZR—fraction of neutrals obtained in the present study; FAZR—fraction of alkaloids obtained in the present study; * Chi-square test; *p* < 0.05 was considered statistically different.

**Table 2 molecules-28-05336-t002:** Prediction of toxicity parameters of molecules isolated from *Z. rhoifolium* for algae and fish.

Mol.	Algae	Daphnia	Fish		Ames	*Carcinogenicity*		Herg *Inhibition*
			Medaka	Minnow		Mouse	Rat	
**1**	T	T	VT	VT	NM	N	P	+
**2**	T	T	VT	VT	M	N	P	+
**3**	T	T	VT	VT	M	N	N	+
**4**	T	T	VT	VT	M	N	P	+
**5**	T	T	VT	VT	NM	N	N	+
**6**	T	T	VT	VT	M	P	N	+
**7**	T	NT	VT	VT	NM	N	N	+
**8**	T	T	VT	VT	M	P	N	+
**9**	T	T	VT	VT	M	N	P	+
**10**	T	T	T	VT	NM	P	N	+
**11**	T	T	VT	VT	NM	P	N	+
**12**	T	T	VT	VT	NM	N	N	+
**13**	T	T	VT	VT	NM	N	N	+
**14**	T	T	VT	VT	NM	P	N	+
**15**	T	T	VT	VT	NM	N	N	+
**16**	T	T	T	VT	NM	N	N	+
**17**	T	T	VT	VT	NM	N	P	+
**18**	T	T	VT	VT	M	N	P	+
**19**	T	T	VT	VT	M	P	N	+
**20**	T	T	VT	VT	M	N	N	+
**21**	T	T	VT	VT	M	N	N	+
**22**	T	T	T	VT	M	N	P	+
**23**	NT	T	VT	VT	NM	N	P	+
**24**	T	T	VT	VT	NM	P	P	+
**25**	T	T	VT	VT	NM	N	P	+
**26**	T	T	VT	VT	M	P	N	+
**27**	T	VT	VT	VT	M	N	P	+
**28**	T	T	VT	VT	M	P	N	+
**29**	T	T	VT	VT	M	N	N	+

Legend: NT: non-toxic; VT: very toxic; T: toxic; M: mutagenic; NM: non-mutagenic; P: positive; N: negative; +: inhibiting action; **1**—rhoifoline A, **2**—rhoifoline B, **3**—dihydronitidine, **4**—decarine, **5**— (+/−)-6-Acetonyldihydrochelerythrine, **6**—1-methoxy-7,8-dehydrorutaceacarpine, **7**—magnoflorine, **8**—5-methoxy-canthin-6-one, **9**—zanthoxyline, **10**—nitidine, **11**—avicine, **12**—6-acetonyldihydroavicine; **13**—columbamine, **14**—chelerythrine, **15**—O-methylcapaurine, **16**—syncarpamide, **17**—bocconoline, **18**—nornitidine, **19**—dictamine, **20**—dihydroavicine, **21**—oxynitidine, **22**—norfagaronine, **23**—bis[6-(5,6-dihydrochelerythrinyl)], **24**—dectamnine, **25**—skimmianine, **26**—pelitorine, **27**—Z-dimethylrhoifolinate, **28**—canthin-6-one, and **29**—quinazoline-6-carboxylic acid.

**Table 3 molecules-28-05336-t003:** Prediction of oral toxicity parameters of molecules isolated from *Z. rhoifolium*.

Molecules	LD_50_ (mg/Kg)	Classification	Toxic Effect
**1**	1250	N	I
**2**	1250	N	I
**3**	167	T	I
**4**	1000	N	I
**5**	778	N	I
**6**	445	N	I
**7**	401	N	I
**8**	1000	N	I
**9**	778	N	I
**10**	940	NT	NTE
**11**	347	N	I
**12**	1000	N	I
**13**	200	T	I
**14**	778	N	I
**15**	580	N	I
**16**	650	N	I
**17**	296	T	I
**18**	640	N	I
**19**	1000	N	NTE
**20**	167	T	I
**21**	167	T	I
**22**	2300	P	I
**23**	2000	N	I
**24**	1000	N	NTE
**25**	1000	N	I
**26**	700	N	I
**27**	305	N	NTE
**28**	1200	N	NTE
**29**	3000	P	NTE

LD_50_: lethal dose 50%; NT: non-toxic; P: may be harmful if ingested; T: toxic if ingested; I: immunotoxic; Cit: cytotoxic; NTE: no toxic effect; **1**—rhoifoline A, **2**—rhoifoline B, **3**—dihydronitidine, **4**—decarine, **5**— (+/−)-6-Acetonyldihydrochelerythrine, **6**—1-methoxy-7,8-dehydrorutaceacarpine, **7**—magnoflorine, **8**—5-methoxy-canthin-6-one, **9**—zanthoxyline, **10**—nitidine, **11**—avicine, **12**—6-acetonyldihydroavicine; **13**—columbamine, **14**—chelerythrine, **15**—O-methylcapaurine, **16**—syncarpamide, **17**—bocconoline, **18**—nornitidine, **19**—dictamine, **20**—dihydroavicine, **21**—oxynitidine, **22**—norfagaronine, **23**—bis[6-(5,6-dihydrochelerythrinyl)], **24**—dectamnine, **25**—skimmianine, **26**—pelitorine, **27**—Z-dimethylrhoifolinate, **28**—canthin-6-one, and **29**—quinazoline-6-carboxylic acid.

**Table 4 molecules-28-05336-t004:** Free energy (ΔG), inhibition constant (Ki), and connections established between test samples and topoisomerase II.

Binder	ΔG Kcal/mol	Ki µM	Hydrogen Bridges	Hydrophobic Bonds	π-Bond
Rhoifoline A	4.58	439.97	Lys37	His42, Leu45, Tyr34, Gln35	Ile33
Rhoifoline B	5.4	110.97	Gln35	Ile33, Lys36	Tyr34, Lys37
Decarine	5.0	217.56	Ile33	Tyr34, Gln35, Lys37, Gln39, Thr38	His42, Gln35, Lys37
Zanthoxyline	5.08	189.94	Ile33	Lys36, Leu45, His42	Gln35, Tyr34, Lys37

## Data Availability

Data are available from the corresponding author upon request.

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
