# Peer review of "Cytotoxicity and Genotoxicity Evaluation of Zanthoxylum rhoifolium Lam and In Silico Studies of Its Alkaloids"

_molecules, 2023, doi:10.3390/molecules28145336_

Round 1
Reviewer 1 Report
Introduction should be expanded. The importance of the topic should be clearly described.
Author Response
Q. Introduction should be expanded. The importance of the topic should be clearly described.
The intro has been changed
Reviewer 2 Report
In this study, the authors investigated the cytotoxicity and genotoxicity of extracts and fractions from Z. rhoifolium using HepG2 cells. Although these samples did not induce cytotoxic effects, they exhibited genotoxicity. From the in silico assay involving molecular docking on TopoII, the authors hypothesized that alkaloids with a benzophenanthridine nucleus are involved in this toxicity.
The authors should include previous research or experimental information in the introduction. In the introduction, the authors mention, "there is a lack of studies assessing the toxic potential of this plant and whether this possible toxicity is related to the alkaloids present in the species" (lines 64-65). However, as described in the discussion section, cytotoxicity of the extracts for Vero cells and gastric cells has been previously reported. Furthermore, the manuscript lacks information on toxicities, making it unclear to readers why genotoxicity was investigated. These essential details should be included in the introduction; otherwise, readers may misunderstand the aim, concept, and novelty of the present study.
In the references section, the authors cited several master's theses, such as refs. [9], [12], [15], [30], and [32]. Can readers access these references to understand the present study? Is it appropriate to include these master's theses in the reference list of a peer-reviewed article?
The findings of the present study are only applicable to HepG2 cells, as the authors acknowledged varying results in different cell lines in the manuscript. The authors should discuss the differences between HepG2 and other cell lines in the context of research limitations.
Author Response
Q. The authors should include previous research or experimental information in the introduction. In the introduction, the authors mention, "there is a lack of studies assessing the toxic potential of this plant and whether this possible toxicity is related to the alkaloids present in the species" (lines 64-65). However, as described in the discussion section, cytotoxicity of the extracts for Vero cells and gastric cells has been previously reported. Furthermore, the manuscript lacks information on toxicities, making it unclear to readers why genotoxicity was investigated. These essential details should be included in the introduction; otherwise, readers may misunderstand the aim, concept, and novelty of the present study.
Changes were made to the body of the text to address the issues raised.
Q. In the references section, the authors cited several master's theses, such as refs. [9], [12], [15], [30], and [32]. Can readers access these references to understand the present study? Is it appropriate to include these masters theses in the reference list of a peer-reviewed article?
In Brazil, all dissertations are freely available at the Brazilian Digital Library of Theses and Dissertations. However, we changed the references cited for peer-reviewed articles.
Q. The findings of the present study are only applicable to HepG2 cells, as the authors acknowledged varying results in different cell lines in the manuscript. The authors should discuss the differences between HepG2 and other cell lines in the context of research limitations.
The speech was held.
Reviewer 3 Report
Dear Authors:
Before my suggestions I would like to thank you for your work, it certainly leads to the discussion of genotoxicity and cytotoxicity of medicinal plant extracts. I consider your text to be appropriate, but allow me to make the following respectful suggestions: 1. I suggest that the title be more descriptive and return to some of its main findings, for example "Low cytotoxicity and moderate genotoxicity of extracts of Zanthoxylum rhoifolium Lam: in vitro, in vivo and in silico assays". 2. It needs to expand its discussion on genotoxicity. I suggest you review the levels of spontaneous aberrations in the Allium strain trial so that you can qualify your discussion of it. 3. In addition, I suggest organizing the discussion of the results by topic. For example, you can first address cytotoxicity in all three scenarios (in vitro, in vivo and in silico) and then discuss genotoxicity. Finally, it is clear what was the most valuable of his findings.
Author Response
Q. 1. I suggest that the title be more descriptive and return to some of its main findings, for example "Low cytotoxicity and moderate genotoxicity of extracts of Zanthoxylum rhoifolium Lam: in vitro, in vivo and in silico assays". 2. It needs to expand its discussion on genotoxicity. I suggest you review the levels of spontaneous aberrations in the Allium strain trial so that you can qualify your discussion of it. 3. In addition, I suggest organizing the discussion of the results by topic. For example, you can first address cytotoxicity in all three scenarios (in vitro, in vivo and in silico) and then discuss genotoxicity. Finally, it is clear what was the most valuable of his findings.
Greetings, dear reviewer.
Thank you for your comments and we reply:
1) We don't think that the title should deliver the "gold" before its time, we think that
this way, we instigate the reading of the text.
2) Changes have been made.
3) Changes have been made.
Once again, thank you very much for your encouragement and for your contributions to our work.
Round 2
Reviewer 1 Report
The authors made required changes and so expanded the article.
I feel qualified and able to assess the quality of English.
Author Response
Dear reviewer, an English review has been carried out.